# LSTOD: Latent Spatial-Temporal Origin-Destination prediction model and its applications in ride-sharing platforms

## Abstract

Origin-Destination (OD) flow data is an important instrument in transportation studies. Precise prediction of customer demands from each original location to a destination given a series of previous snapshots helps ride-sharing platforms to better understand their market mechanism. However, most existing prediction methods ignore the network structure of OD flow data and fail to utilize the topological dependencies among related OD pairs. In this paper, we propose a latent spatial-temporal origin-destination (LSTOD) model, with a novel convolutional neural network (CNN) filter to learn the spatial features of OD pairs from a graph perspective and an attention structure to capture their long-term periodicity. Experiments on a real customer request dataset with available OD information from a ride-sharing platform demonstrate the advantage of LSTOD in achieving at least 7.94% improvement in prediction accuracy over the second best model.

## 1 Introduction

Spatial-temporal prediction of large-scale network-based OD flow data plays an important role in traffic flow control, urban routes planning, infrastructure construction, and the policy design of ride-sharing platforms, among others. On ride-sharing platforms, customers keep sending requests with origins and destinations at each moment. Knowing the exact original location and destination of each future trip allows platforms to prepare sufficient supplies in advance to optimize resource utilization and improve users' experience. Given the destinations of prospective demands, platforms can predict the number of drivers transferring from busy to idle status. Prediction of dynamic demand flow data helps ride-sharing platforms to design better order dispatch and fleet management policies for achieving the demand-supply equilibrium as well as decreased passenger waiting times and increased driver serving rates.

Many efforts have been devoted to developing traffic flow prediction models in the past few decades. Before the rise of deep learning, traditional statistical and machine learning approaches dominate this field (Li et al., 2012; Lippi et al., 2013; Moreira-Matias et al., 2013; Shekhar & Williams, 2007; Idé & Sugiyama, 2011; Zheng & Ni, 2013). Most of these models are linear and thus ignore some important non-linear correlations among the OD flows. Some other methods (Kwon & Murphy, 2000; Yang et al., 2013) further use additional manually extracted external features, but they fail to automatically extract the spatial representation of OD data. Moreover, they roughly combine the spatial and temporal features when fitting the prediction model instead of dynamically modelling their interactions.

The development of deep learning technologies brings a significant improvement of OD flow prediction by extracting non-linear latent structures that cannot be easily covered by feature engineering. (Xingjian et al., 2015; Ke et al., 2017; Zhou et al., 2018). Zhang et al. (2016; 2017) modeled the whole city are as an entire image and employed residual neural network to capture temporal closeness. Ma et al. (2017) and Yu et al. (2017) also learned traffic as images but they used LSTM instead to obtain the temporal dependency. Yao et al. (2018b) proposed a Deep Multi-View Spatial-Temporal Network (DMVST-Net) framework to model both spatial and temporal relationships. However, using standard convolution filters suffers from the problem that some OD flows covered by a receptive field of regular CNNs are not spatially important. Graph-based neural net-

works (GNN) (Kipf & Welling, 2016; Defferrard et al., 2016; Veličković et al., 2017) are proved to be powerful tools in modelling spatial-temporal network structures (Yu et al., 2018; Li et al., 2017). However, none of these frameworks are directly applicable here since both the historical observations and responses to predict are vertex-level variables. On the contrary, the OD flows we discuss in this paper are generated in the edge space by our definition.

The aim of this paper is to introduce a hierarchical Latent Spatial-Temporal Origin-Destination (LSTOD) prediction model to jointly extract the complex spatial-temporal features of OD data by using some well-designed CNN-based architectures. Instead of modelling the dynamic OD networks as a sequence of images and applying standard convolution filters to capture their spatial information, we introduce a novel Vertex Adjacent Convolution Network (VACN) that uses an irregular convolution filter to cover the most related OD flows that share common vertecies with the target one. The OD flows connected by common starting and/or ending vertexes, which may fall into different regions of the flow map, can be spatially correlated and topologically connected. Moreover, for most ride-sharing platforms, a passenger is more likely to send a new request from the location where his/her last trip ends in. To learn such sequential dependency, we introduce a temporal gated CNN (TGCNN) (Yu et al., 2018) and integrate it with VACN by using the sandwich-structured ST-conv block in order to collectively catch the evolutionary mechanism of dynamic OD flow systems. A periodically shifted attention mechanism is also used to capture the shift in the long-term periodicity. Finally, the combined short-term and long-term representations are fed into the final prediction layer to complete the architecture. Our contributions are summarized as follow:

- To the best of our knowledge, it is the first time that we propose purely convolutional structures to learn both short-term and long-term spatio-temporal features simultaneously from dynamic origin-destination flow data.

- We propose a novel VACN architecture to capture the graph-based semantic connections and functional similarities among correlated OD flows by modeling each OD flow map as an adjacency matrix.

- We design a periodically shift attention mechanism to model the long-term periodicity when using convolutional architecture TGCNN in learning temporal features.

- Experimental results on two real customer demand data sets obtained from a ride-sharing platform demonstrate that LSTOD outperforms many state-of-the-art methods in OD flow prediction, with 7.94% to 15.14% improvement of testing RMSE.

## 2 DEFINITIONS AND PROBLEM STATEMENT

For a given urban area, we observe a sequence of adjacency matrices representing the OD flow maps defined on a fixed vertex set $V$, which indicates the $N$ selected sub-regions from this area. We let $\mathbb{V} = \{v_1, v_2, \ldots, v_N\}$ denote the vertex set with $v_i$ being the $i$-th sub-region. The shape of each grid $v_i$ could be either rectangles, hexagons or irregular sub-regions. We define the dynamic OD flow maps as $\{O_{d,t}\}$, where $d = 1, \ldots, D$ and $t = 1, \ldots, T$ represent the day and time indexes, respectively. For each snapshot $O_{d,t} = (o_{d,t}^{ij}) \in R^{N \times N}$, the edge weight $o_{d,t}^{ij}$ at row $i$ and column $j$ denotes the flow amount from node $v_i$ to node $v_j$ at time $t$ of day $d$. A larger edge weight $o_{d,t}^{ij}$ is equivalent to a strong connection between nodes $v_i$ and $v_j$. The $O_{d,t}$s' are asymmetric since all the included OD flows are directed. Specifically, $o_{d,t}^{ij} = 0$ if there is no demand from $v_i$ to $v_j$ within the $t$-th time interval of day $d$.

The goal of our prediction problem is to predict the snapshot $O_{d,t+j} \in R^{N \times N}$ in the future time window $(t + j)$ of day $d$ given previously observed data, including both short-term and long-term historical information. The short-term input data consists of the last $p_1$ timestamps from $t + 1 - p_1$ to $t$, denoted by $\mathbb{X}_1 = \{O_{d,t+1-p_1}, O_{d,t+1-p_1+1}, \ldots, O_{d,t}\}$. The long-term input data is made up of $q$ time series $\{O_{d-\varphi,t+j-(p_2-1)/2}, \ldots, O_{d-\varphi,t+j+(p_2-1)/2}\}$ of length $p_2$ for each previous day $(d - \varphi)$ with the predicted time index $(t + j)$ in the middle for $\varphi = 1, \ldots, q$. We let $\mathbb{X}_2 = \{O_{d-q,t+j-(p_2-1)/2}, \ldots, O_{d-q,t+j+(p_2-1)/2}, \ldots, O_{d-1,t+j-(p_2-1)/2}, \ldots, O_{d-1,t+j+(p_2-1)/2}\}$ denote the entire long-term data. Increasing $p_1$ and $p_2$ leads to higher prediction accuracy, but more training time.

We reformulate the set of short-term OD networks $\mathbb{X}_1$ into a 4D tensor $\mathbf{X}_1 \in R^{N \times N \times p_1 \times 1}$ and concatenate the long-term snapshots $\mathbb{X}_2$ into a 5D tensor $\mathbf{X}_2 = (\mathbf{X}_{2,d-1}, \ldots, \mathbf{X}_{2,d-q} \in R^{q \times N \times N \times p_2 \times 1}$. Each $\mathbf{X}_{2,d-\varphi} \in R^{N \times N \times p_2 \times 1}$ for day $d - \varphi$ is a 4D tensor for $\varphi = 1, \ldots, q$. Therefore, we can finally define our latent prediction problem as follows:

$$o_{d,t+j} = F(\mathbf{X}_1, \mathbf{X}_2), \tag{1}$$

where $F(\cdot, \cdot)$ represents the LSTOD model, which captures the network structures of OD flow data as well as the temporal dependencies in multiple scales. A notation table is attached in the appendix.

## 3 LSTOD FRAMEWORK

In this section, we describe the details of our proposed LSTOD prediction model. See Figure 1 for the architecture of LSTOD. The four major novelties and functionalities of LSTOD model include

- an end-to-end framework LSTOD constructed by all kinds of CNN modules to process dynamic OD flow maps and build spatio-temporal prediction models;
- a novel multi-layer architecture VACN to extract the network patterns of OD flow maps by propagating through edge connections, which can not be covered by traditional CNNs;
- a special module ST-Conv block used to combine VACN and gated temporal convolutions to coherently learning the essential spatio-temporal representations;
- a periodically shifted attention mechanism which is well designed for the purely convolutional ST-Conv blocks to efficiently utilize the long-term information by measuring its similarities with short-term data.

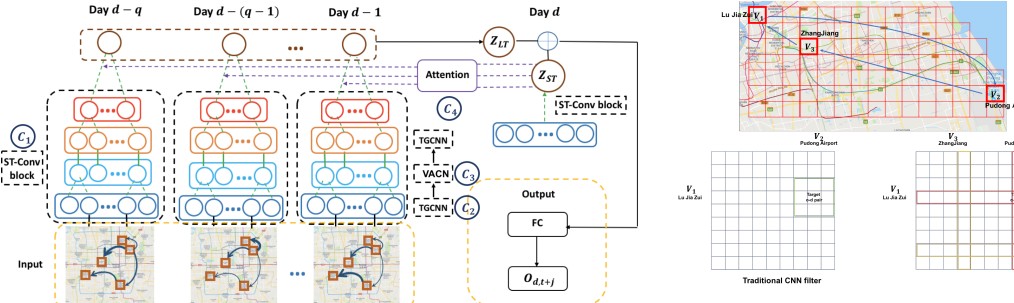

Figure 1: The architecture of LSTOD consisting of (C1) ST-Conv blocks, (C2) TGCNN, (C3) VACN, and (C4) attention.

Figure 2: An example to show how standard CNN fails to capture the network structure of OD flows

### 3.1 SPATIAL ADJACENT CONVOLUTION NETWORK

Before introducing the detailed architecture of VACN, we want to discuss why directly applying standard CNNs to the OD flow map $\boldsymbol{O}_{d,t}$ may disregard the connections between neighboring OD flows in the graph space first. Figure 2 demonstrates that it fails to capture enough semantic information using the real-world example of ride demands. For the OD flow starting from $v_1$ to $v_2$, as illustrated in the upper sub-figure, the most related OD flows should be those with either origin or destination being $v_1$ or $v_2$ in the past few timestamps. A certain part of the travel requests from $v_1$ to $v_2$ can be matched with some historical finished trips from a third-party location to $V_1$ by the same group of people, for example a trip from $v_3$ to $v_1$. However, as the lower-left sub-figure illustrates, some of the OD flows covered by a single CNN filter (the green square) such as the four corners of the kernel window may be topologically far away from the target one in the graph.

More generally, let's consider a target OD flow $o_{d,t}^{ij}$ in a map $\boldsymbol{O}_{d,t}$. Most of the components covered by a standard CNN of size $q \times q$ with $o_{d,t}^{ij}$ in the middle such as $o_{d,t}^{(i+1)(j+1)}$ are less correlated than some out of the kernel window, for example $o_{d,t}^{ki}$ when $|k - j| > (q + 1)/2$. Moreover, if we change

the order of the $N$ vertexes in $\boldsymbol{O}_{d,t}$, then the network structure is unchanged, but a different set of OD flows is covered by the receptive field of the same size with the center being $o_{d,t}^{ij}$. As shown in the lower right sub-figure of Figure 2, OD flows with either origin or destination being $v_i$ or $v_j$, covered by the red and yellow kernel windows, are considered to be the most related ones for $o_{d,t}^{ij}$ in row $i$ and column $j$.

To better understand the differences between our proposed VACN over graph edges with those vertex-based convolutions such as GCN (Kipf & Welling, 2016) or GAT (Veličković et al., 2017), we introduce the concept of line graphs $L(G)$ (Godsil & Royle, 2013). Each node in $L(G)$ corresponds to an edge in $G$, while each individual edge in $L(G)$ can be mapped to a pair of edges in $G$ that connect to a joint vertex. $L(G)$ contains $\frac{1}{2}[N^3 - N(N-1)]$ while there are only $\frac{1}{2}N(N-1)$ for $G$. Thus, applying GNNs over $L(G)$ is memory intensive. Our edge-based convolution operations VACN is much more efficient.

Formally, every VACN layer takes a 3D snapshot $\mathbf{A}_{d,t} = \{\boldsymbol{a}_{d,t}^{ij}\} \in R^{N \times N \times m_0}$ consist of $m_0$ feature maps as the input with each $\boldsymbol{a}_{d,t}^{ij} \in R^{m_0}$ representing a feature vector at the edge from $v_i$ to $v_j$. The learned representation of each target edge is defined as the weighted sum of those from the same row or column in the adjacency matrix, and those from the row or column in the transposed adjacency matrix. The output of the layer-wise VACN propagation for the OD flow from $v_i$ to $v_j$ is defined as follows:

$$\mathcal{F}\{\sum_{n=1}^{N} \boldsymbol{R}_1^n \boldsymbol{a}_{d,t}^{in} + \boldsymbol{C}_1^n \boldsymbol{a}_{d,t}^{nj} + \boldsymbol{R}_2^n \boldsymbol{a}_{d,t}^{ni} + \boldsymbol{C}_2^n \boldsymbol{a}_{d,t}^{jn}\} \in R^{m_1}, \tag{2}$$

where $[\boldsymbol{R}_1^n, \boldsymbol{R}_2^n, \boldsymbol{C}_1^n, \boldsymbol{C}_2^n] = \boldsymbol{W}^n \in R^{4m_1 \times m_0}$ such that $[\boldsymbol{W}^1, \ldots, \boldsymbol{W}^N] \in R^{4m_1 \times m_0 \times N}$ are the weights to learn. $\mathcal{F}(\cdot)$ represents an elementwise activation function, such as ReLU(x) = $\max(0, x)$. The first part of (2) works by summing up the feature values of OD flows having either the same origin or destination with the target OD flow. The second part covers another set of OD flows that either start at $v_j$ or end at $v_i$. Similar to standard CNNs, different OD flows are treated with unequal importance by VACN and edges more related to the target one are assigned higher weights $\boldsymbol{R}_1^n, \boldsymbol{R}_2^n, \boldsymbol{C}_1^n$ or $\boldsymbol{C}_2^n$. We denote the overall output of a multi-layer VACN as $\mathbf{Y}_{d,t} \in R^{N \times N \times m_s}$ at time $(d,t)$. Figure 4 in the appendix more clearly illustrates the architecture of VACN.

## 3.2 TEMPORAL GATED CNN

We use temporal gated CNN (TGCNN) (Yu et al., 2018) instead of RNN-based architectures such as LSTMs to capture the temporal representation, which makes our LSTOD a pure convolutional architecture. RNNs suffer from the problem of lower training efficiency, gradient instability, and time-consuming convergence. Moreover, the high dimension of the spatial representations captured by VACN and a potential long temporal sequence length make RNNs notoriously difficult to train. The CNNs is more flexible in handling various data structures and allows parallel computations to increase training speed.

TGCNN consists of two parts including one being a 3D convolution kernel applied to the spatial representations of all the $N^2$ OD flows along the time axis and the other being a gated linear unit (GLU) as the gate mechanism. By employing VACN at each of $r$ successive timeslots, we can build a 4D tensor $\mathbf{Y} = (\mathbf{Y}_{d,t}) \in R^{N \times N \times r \times m_s}$ which is then fed into the TGCNN operator:

$$G *_\gamma \mathbf{Y} = \mathbf{Y}_1 \odot \sigma(\mathbf{Y}_2) \in R^{N \times N \times (r-K+1) \times m_t}, \tag{3}$$

where $G*_\gamma$ represents the TGCNN kernel and $\gamma$ includes all the related parameters to learn. $2m_t$ 3D convolutional kernels of size $(1 \times 1 \times K)$ with zero paddings map the input $\mathbf{Y}$ into a single output $[\mathbf{Y}_1\ \mathbf{Y}_2] \in R^{N \times N \times (r-K+1) \times 2m_t}$, which is split in half to obtain $\mathbf{Y}_1$ and $\mathbf{Y}_2$ with the same number of feature channels. $\odot$ here denotes the element-wise Hadamard product.

## 3.3 ST-CONV BLOCKS

We use the spatial-temporal convolutional block (ST-conv block) to jointly capture the spatial-temporal features of OD flow data, which has a 'sandwich'-structure architecture with a multi-layer VACN operator in the middle connecting the two TGCNNs. The use of ST-Conv blocks have two

major advantages. First, the block can be stacked or extended based on the dimension and characteristic of the spatio-temporal input data. Second, a temporal operation is applied before extracting the spatial information, which greatly reduces its computation complexity and memory consumption.

Both the input and output of each individual ST-Conv block are 4D tensors. For the input of the $l$-th block $\mathbf{Z}^{l-1} \in R^{N \times N \times r_{l-1} \times c_{l-1}}$ ($\mathbf{Z}^0$ is the original the OD flow data with $c_0 = 1$), the output is computed as follows:

$$\mathbf{Z}^l = G_1 *_{\gamma_1^l} [S *_{\theta^l} \{G_0 *_{\gamma_0^l} \mathbf{Z}^{l-1}\}], \tag{4}$$

where $G_1*_{\gamma_1^l}$ and $G_0*_{\gamma_0^l}$ are two TGCNN kernels and $S*_{\theta^l}$ is a multi-layer VACN operator being applied to each timestamp. The $G_1$ and $G_2$ operators from all the stacked ST-Conv blocks employ the same kernel sizes, which are $(1 \times 1 \times K_1)$ and $(1 \times 1 \times K_2)$, respectively. Thus, we have $r_l = r_{l-1} - (K_1 + K_2 - 2)$. After applying $(r_0 - 1)/(K_0 + K_1 - 2)$ ST-Conv blocks to the input $\mathbf{Z}^0$, the temporal length is reduced from $r_0$ to 1.

When input being the short-term OD flow data $\mathbf{Z}^0 = \mathbf{X}_1 \in R^{N \times N \times p_1 \times 1}$, we use $L_0 = (p_1 - 1)/(K_{ST}^0 + K_{ST}^1 - 2)$ blocks to obtain the spatial-temporal representations $\mathbf{Z}_{ST} = f(\mathbf{Z}^{L_0}) \in R^{N \times N \times c_{ST}}$. $f$ here squeezes the captured 4D output into a 3D tensor by dropping the temporal axis. The kernel sizes of each $G_1$ and $G_0$ are $(1 \times 1 \times K_0^{ST})$ and $(1 \times 1 \times K_1^{ST})$, respectively. The detailed propagation of the $l$-th ST-Conv block is defined as

$$\mathbf{Z}^l = G_1 *_{\gamma_{ST}^{l1}} [S *_{\theta_{ST}^l} \{G_0 *_{\gamma_{ST}^{l0}} \mathbf{Z}^{l-1}\}], \tag{5}$$

## 3.4 Periodically Shifted Attention Mechanism

In addition to capturing the the spatial-temporal features from short-term OD flow data $\mathbb{X}_1$, we also take into account the long-term temporal periodicity due to the potential day-wise cycling patterns insides the OD flow data, decided by customer's travelling schedule and the city's traffic conditions. Directly applying ST-Con blocks to an extremely long OD sequence which covers previous few days or weeks is computationally expensive. Only a small set of timestamps from each previous day is necessary to capture the long-term periodicity. As mentioned, we pick $p_2$ time intervals for each day $d - \varphi$ when predicting the time window $(d, t + j)$ considering the non-strict long-term periodicity. This slight time shifting may be caused by unstable traffic peaks, holidays and extreme weather conditions among different days.

Inspired by the recently widely used attention mechanisms (Xu et al., 2015; Yao et al., 2018a; Liang et al., 2018) in spatial-temporal prediction problems, we propose a modified periodically shifted attention to work for the CNN-based ST-Conv blocks here. Different from Yao et al. (2018a) that measures the similarity between hidden units of LSTMs, the attention here is built on the intermediate outputs of TGCNNs where the concatenations are then fed into a new set of ST-Conv blocks. For each day $(d - \varphi)$, we apply a series of $L_1$ ST-Conv blocks to the day-level $p_2$-length sequential OD flow data $\mathbf{X}_{2,d-\varphi}$ and reduce the temporal length from $p_2$ to $n_{LT}^0$. Each block contains two TGCNN layers with the same kernel size $1 \times 1 \times K_{LT}^0$, such that

$$p_2 - n_{LT}^0 = L_1(2K_{LT}^0 - 2) \tag{6}$$

and the propagation rule of the $l$-th ST-Conv blocks is defined as:

$$\mathbf{Z}_{d-\varphi}^l = G_1 *_{\gamma_{LT}^{l1}} [S *_{\theta_{LT}^l} \{G_0 *_{\gamma_{LT}^{l0}} \mathbf{Z}_{d-\varphi}^{l-1}\}] \tag{7}$$

with $\mathbf{Z}_{d-\varphi}^n$ and $\mathbf{Z}_{d-\varphi}^{n+1}$ representing the input and output of the $l$-the block.

We denote the day-level representations of day $(d - \varphi)$ captured by the $L_1$ ST-Conv blocks above as $\tilde{\mathbf{Z}}_{d-\varphi}$ where each $\tilde{z}_{d-\varphi,\phi}^{ij}$ represents the $\phi$-th element along the time axis for the OD flow from $v_i$ to $v_j$. On the other hand, we let $z_{ST}^{ij}$ be the learned short-term representation at the OD flow from $v_i$ to $v_j$. Subsequently, a re-weighted day-level output $z_{d-\varphi}^{ij}$ can be obtained by summing up $\tilde{z}_{d-\varphi,\phi}^{ij}$'s using weights $\beta_{d-\varphi,\phi}^{ij}$ which measure their similarities with the short-term representation $z_{ST}^{ij}$:

$$z_{d-\varphi}^{ij} = \sum_{\phi=1}^{n_{LT}^0} \beta_{d-\varphi,\phi}^{ij} \tilde{z}_{d-\varphi,\phi}^{ij}, \tag{8}$$

where $\beta_{d-\varphi,\phi}^{ij}$ quantifies the similarity between $\tilde{z}_{d-\varphi,\phi}^{ij}$ and $z_{ST}^{ij}$ based on a score function $\text{score}(\tilde{z}_{d-\varphi,\phi}^{ij}, z_{ST}^{ij})$, which is defined as:

$$\beta_{d-\varphi,\phi}^{ij} = \frac{\exp(\text{score}(\tilde{z}_{d-\varphi,\phi}^{ij}, z_{ST}^{ij}))}{\sum_{\phi'} \exp(\text{score}(\tilde{z}_{d-\varphi,\phi'}^{ij}, z_{ST}^{ij}))}. \tag{9}$$

Moreover, $\text{score}(\tilde{z}_{d-\varphi,\phi}^{ij}, z_{ST}^{ij})$ is defined as

$$v_\phi^T \tanh(\boldsymbol{W}_1 \tilde{z}_{d-\varphi,\phi}^{ij} + \boldsymbol{W}_2 z_{ST}^{ij} + b_s), \tag{10}$$

where $\boldsymbol{W}_1, \boldsymbol{W}_2$ and $v_\phi$ are learned projection matrices. $b_s$ is the added bias term. By assuming that $\mathbf{Z}_{d-\varphi} = (z_{d-\varphi}^{ij})$ denotes the day-level output for all the $N^2$ entries after re-weighting, we can build a new day-wise time series $\mathbf{Z}_{LT}^0$ of length $q$ by concatenating all the $\mathbf{Z}_{d-\varphi}$'s along an additional axis in the third dimension as

$$\mathbf{Z}_{LT}^0 = \text{Concat}_{\varphi=q}^1 \mathbf{Z}_{d-\varphi} \tag{11}$$

to build a new 4D day-wise time series $\mathbf{Z}_{LT}^0$. and finally apply another set of ST-Conv blocks to it to obtain the long-term spatial-temporal representations, which is denoted by $\mathbf{Z}_{LT} \in R^{N \times N \times c_{LT}}$. $c_{LT}$ is the number of feature channels.

We concatenate the short-term and long-term spatial-temporal representations $\mathbf{Z}_{ST}$ and $\mathbf{Z}_{LT}$ together along the feature axis as $\mathbf{Z} = \mathbf{Z}_{ST} \oplus \mathbf{Z}_{LT} \in R^{N \times N \times \mathcal{C}}$, where $\mathcal{C} = c_{ST} + c_{LT}$. Then, $\mathbf{Z}$ is modified to a 2D tensor $\boldsymbol{Z} \in R^{N^2 \times \mathcal{C}}$ by flattening the first two dimensions while keeping the third one. We apply a fully connected layer to the $\mathcal{C}$ feature channels together with an element-wise non-linear sigmoid function to get the final predictions for all the $N^2$ OD flows.

### 3.5 DATA PROCESSING AND TRAINING

We normalize the original OD flow data in the training set to $(0, 1)$ by Max-Min normalization and use 'sigmoid' activation for the final prediction layer to ensure that all predictions fall into $(0, 1)$. The upper and lower bounds are saved and used to denormalize the predictions of testing data to get the actual flow volumes.

We use $L_2$ loss to build the objective loss during the training. The model is optimized via Back-propagation Through Time (BPTT) and Adam (Kingma & Ba, 2014). The whole architecture of our model is realized using Tensorflow (Abadi et al., 2016) and Keras (Chollet et al., 2015). All experiments were run on a cluster with one NVIDIA 12G-memory Titan GPU.

## 4 EXPERIMENT

In this section, we compare the proposed LSTOD model with some state-of-the-art approaches for latent traffic flow predictions. All compared methods are classified into traditional statistical methods and deep-learning based approaches. We use the demand flow data collected by a ride-sharing platform to examine the finite sample performance of OD flow predictions for each method.

### 4.1 DATASET DESCRIPTION

We employ a large-scale demand dataset obtained from a large-scale ride-sharing platform to do all the experiments. The dataset contains all customer requests received by the platform from 04/01/2018 to 06/30/2018 in two big cities A and B. Within each urban area, $N = 50$ hexagonal regions with the largest customer demands are selected to build the in total $N^2 = 2500$ OD flows. Since one-layer VACN has a computation complexity $O(N)$ at each of the $N^2$ entries (globally $O(N^3)$), the memory consumption highly increases as $N$ gets bigger. Considering the computation efficiency and storage limitation, we choose $N = 50$ here which can cover more than 80% of total demands and thus satisfy the operation requirement of the ride-sharing platform.

We split the whole dataset into two parts. The data from 04/01/2018 to 06/16/2018 is used for model training, while the other part from 06/17/2017 to 06/30/2017 (14 days) serves as the testing

set. The first two and half months of OD flow data is further divided in half to the training and validation sets. The size ratio between the two sets is around 4:1. We let 30 min be the length of each timestamp and the value of the OD flow from $v_i$ to $v_j$ is the cumulative number of customer requests. We make predictions for all the $50^2$ OD flows in the incoming 1st, 2nd, and 3rd 30 minutes (i.e. $t+1, t+2, t+3$) by each compared method, given the historical data with varied $(p_1, p_2)$ combinations. For those model settings incorporating long-term information, we trace back $q = 3$ days to capture the time periodicity. We use Rooted Mean Square Error to evaluate the performance of each method:

$$\text{RMSE} = \sqrt{\frac{1}{N^2 * |\mathcal{T}_0|} \sum_{i=1}^{N} \sum_{j=1}^{N} \sum_{(d,t) \in \mathcal{T}_0} (o_{d,t}^{ij} - \widehat{o}_{d,t}^{ij})^2}, \tag{12}$$

$o_{d,t}^{ij}$ and $\widehat{o}_{d,t}^{ij}$ are the true value and prediction at the OD flow from vertex $v_i$ to vertex $v_j$ at time $(d, t)$, respectively. $\mathcal{T}_0$ is the set containing all the predicted time points in the testing data.

## 4.2 COMPARED METHODS

All state-of-the-art methods to be compared are listed as follows, some of which are modified to work for the OD flow data: (i) **Historical average (HA)**: HA predicts the demand amount at each OD flow by the average value of the same day in previous 4 weeks. (ii) **Autoregressive integrated moving average (ARIMA)**, (iii) **Support Vector Machine Regression (SVMR)**, (iv) **Latent Space Model for Road Networks (LSM-RN)** (Deng et al., 2016), (v) **Dense + BiLSTM (DLSTM)** (Altché & de La Fortelle, 2017) and (vi) **Spatiotemporal Recurrent Convolutional Networks (SRCN)** (Yu et al., 2017). We only consider latent models in this paper, that is, no external covariates are allowed, while only the historical OD flow data is used to extract the hidden spatial-temporal features.

## 4.3 PREPROCESSING AND PARAMETERS

We tune the hyperparameters of each compared model to obtain the optimal prediction performance. Specifically, we get $(p^*, d^*, q^*) = (3, 0, 3)$ for ARIMA and $k^* = 15, \gamma^* = 2^{-5}, \lambda^* = 10$ for LSM-RN. The optimal kernel size of the spatial based CNN kernel is $11 \times 11$ in SRCN model.

For fair comparison, we set the length of short-term OD flow sequence to be $p_1 = 9$ (i.e., previous 4.5 hours), $q = 3$ for long-term data which covers the three most recent days, and the length of each day-level time series $p_2 = 5$ to capture the periodicity shifting (one hour before and after the predicted time index). More analysis of how variant $(p_1, p_2)$ combinations may affect the prediction performance of LSTOD will be studied latter.

A two-layer architecture is used by all the deep-learning based methods to extract the spatial patterns inside the OD flow data (L = 2 for both short-term and long-term VACN). We set the filter size of all deep learning layers in both spatial and temporal space to be 64, including the VACNs and TGCNNs in our LSTOD model with $c_{ST} = c_{LT} = 64$.

## 4.4 RESULTS

**Comparison with state-of-the-art methods**. Table 1 summarizes the finite sample performance for all the competitive methods and our LSTOD model in terms of the prediction RMSE on the testing data of city A. We compute the mean, variance, 25% quantile and 75% quantile of the 14 day-wise RMSE on the testing set. LSTOD outperforms all other methods on the testing data with the lowest average day-wise RMSE (2.41/2.55/2.67), achieving (8.02%/7.94%/8.24%) improvement over the second best method 'SRCN'. In general, deep-learning based models perform more stably than traditional methods with smaller variance and narrower confidence intervals. Both 'ARIMA' and 'LSM-RN' perform poorly, even much worse than HA, indicating that they cannot capture enough short-term spatial-temporal features to get the evolution trend of OD flow data. Among the deep learning models, LSTOD can more efficiently control the estimation variance compared to all the others. This demonstrates the advantages of using our spatial-temporal architecture and long-term periodicity mechanism in modelling the dynamic evolution of OD flow networks. The improvement becomes more significant when the time scale increases since the contributions of

Table 1: Comparison with State-of-the-art methods for City A

| Method | City A | | |
|--------|---------|---------|---------|
| | 30 min | 60 min | 90 min |
| HA | | 3.61(3.24/3.77/0.41) | |
| ARIMA | 5.11(4.36/5.82/1.21) | 5.37(4.62/6.11/1.29) | 5.87(5.09/6.65/1.39) |
| LSVR | 3.55(2.55/4.51/2.09) | 3.83(2.64/4.94/2.95) | 4.65(3.42/5.78/3.24) |
| LSM-RN | 5.62(4.61/8.72/1.86) | 6.26(5.41/7.19/1.97) | 6.70(5.71/7.63/2.12) |
| DLSTM | 3.03(2.51/3.50/0.52) | 3.52(2.89/3.03/0.57) | 3.91(3.33/4.48/0.72) |
| SRCN | 2.62(2.09/3.12/0.54) | 2.77(2.14/3.35/0.72) | 2.91(2.18/3.60/0.93) |
| **LSTOD** | **2.41(2.16/2.69/0.17)** | **2.55(2.27/2.68/0.19)** | **2.67(2.35/3.01/0.25)** |

long-term periodicity are more emphasized as the short-term signals getting weaker. The LSTOD performs even better on city B compared to the baseline methods since the long-term periodical pattern in city B may be more significant compared with that in city A. Detailed results about City B are summarized in Table 3 of the appendix.

**Comparison with variants of LSTOD**. Table 2 shows the finite sample performance of our proposed model LSTOD and its different variants based on the demand data from city A. We can see that the complete LSTOD model outperforms the short-term model and the one without using attention mechanisms in terms of smaller means and variances, and narrower confidence intervals. It indicates that the attention design we use can capture the shift of the day-wise periodicity and extract more seasonal patterns to improve prediction accuracy. The left sub-plot of Figure 3 compares the predictions by each model against the true values at two selected OD flows on the last 3 testing days in the 60-minute scales. Two abnormal change points are marked by black circles. The short-term model fails in this case because it ignores the long-term information. The complete LSTOD model outperforms the one without using attention mechanisms since it can better catch the shift of the periodicity. The right sub-plot visualizes the distribution curves of the day-wise RMSEs on the 14 testing days by each of the three compared models. The lighter tail of the red curve demonstrates that the complete LSTOD is more predictive and stable especially for those unusual cases. We do some more experiments to show how different hyperparameter configurations influence the model performance. For more details, please refer to Section E of the appendix.

**VACN VS standard local CNN**. In this experiment, we will show that our proposed VACN outperforms standard CNNs in capturing the hidden network structure of the OD flow data. Given the model setting that $N = 50$ sub-regions of city A are used to build the dynamic OD flow matrices, the number of pixels being covered by VACN at each single snapshot is $50 \times 4 = 200$. For fair comparison, the largest receptive filed of standard CNN should be no bigger than a $15 \times 15$ window, which includes 225 elements each time. We consider five different kernel sizes including $5 \times 5$, $8 \times 8$, $11 \times 11$, $14 \times 14$, and $15 \times 15$. We replace VCAN in our model by standard CNN in order to fairly compare its performance. All hyper-parameters are fixed but only the kernel size of CNNs being changed. Moreover, we only consider the baseline short-term mode of LSTOD model while ignoring the long-term information. As Figure 4 illustrates, standard CNN achieves the best performance with the smallest RMSE $= 2.64$ on testing data with the filter size being $11 \times 11$, which is still higher than that using VACN with RMSE $= 2.54$. Specifically, RMSE increases when the receptive field is larger than $11 \times 11$ since the spatial correlations among the most related OD flows (sharing common origin or destination nodes) are smoothed with the increase in the filter size ($(8 \times 2 - 1)/64 > (14 \times 2 - 1)/196$). This experiment shows that treating the dynamic demand matrix as an image, and applying standard CNN filters does not capture enough spatial correlations among related OD flows without considering their topological connections from the perspective of graphs. For more details, please refer to Figure 4.

As we mentioned above, one-layer VACN has a global computation complexity $O(N^3)$ at each timestamp. For standard CNN, $O(N^2)$ executions are still needed by applying local CNN filters to $N^2$ windows with each of the $N^2$ entries in the middle. Therefore, when the side length of the CNN filter is in the order $O(\sqrt{N})$, the total cost of our VACN and standard CNN are in the same order.

| Method | RMSE | | |
| --- | --- | --- | --- |
| | 30 min | 60 min | 90 min |
| VACN + TGCNN (short-term) | 2.52(2.21/2.90/0.26) | 2.74(2.44/2.81/0.34) | 2.81(2.33/3.25/0.45) |
| LSTOD (no attention) | 2.47(2.14/2.81/0.23) | 2.62(2.31/2.70/0.28) | 2.70(2.34/3.12/0.35) |
| **LSTOD (complete)** | **2.41(2.16/2.69/0.17)** | **2.56(2.27/2.68/0.19)** | **2.67(2.35/3.01/0.25)** |

Table 2: Evaluation of LSTOD and its variants

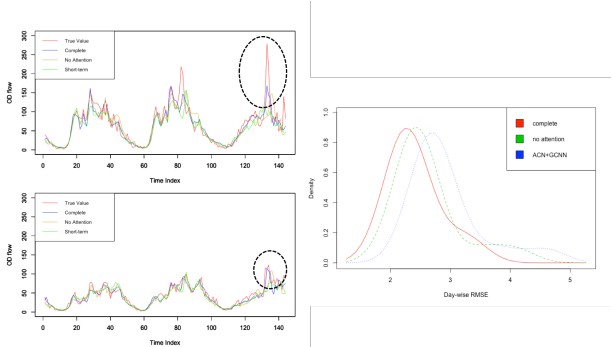
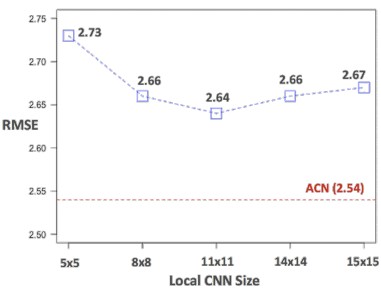

Figure 3: Day-wise RMSE comparison between varied STOD models

Figure 4: RMSE on testing data with respect to ACN and standard CNN using different kernel sizes.

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

## A    NOTATION TABLE

| | |
|---|---|
| $\boldsymbol{O}_{d,t}$ | OD flow maps of day $d$, time $t$ |
| $o_{d,t}^{ij}$ | Flow amount from vertex $v_i$ to $v_j$ at day $d$, time $t$ |
| $\mathbf{X}_1, \mathbf{X}_2$ | Short-term and long-term spatial-temporal OD flow data |
| $\mathbf{X}_{2,d-\varphi}$ | long-term spatial-temporal OD flow data of day $d - \varphi$ |
| $\mathbf{Z}_{d-\varphi}^{n+1}$ | Output of $n$-th ST-Conv block of day $d - \varphi$ |
| $\tilde{\mathbf{Z}}_{d-\varphi}$ | features of day $(d - \varphi)$ captured by ST-Conv blocks |
| $\tilde{z}_{d-\varphi,\phi}^{ij}$ | $\phi$-th element along the time axis of $\tilde{\mathbf{Z}}_{d-\varphi}$ from $v_i$ to $v_j$ |
| $\mathbf{Z}_{d-\varphi}$ | day-level output of day $d - \varphi$ |
| $\mathbf{Z}_{LT}$ | long-term spatial-temporal representations |
| $Z$ | $\mathbf{Z}_{ST} \oplus \mathbf{Z}_{LT}$, combined spatial-temporal representations |

## B    ILLUSTRATION OF VACN

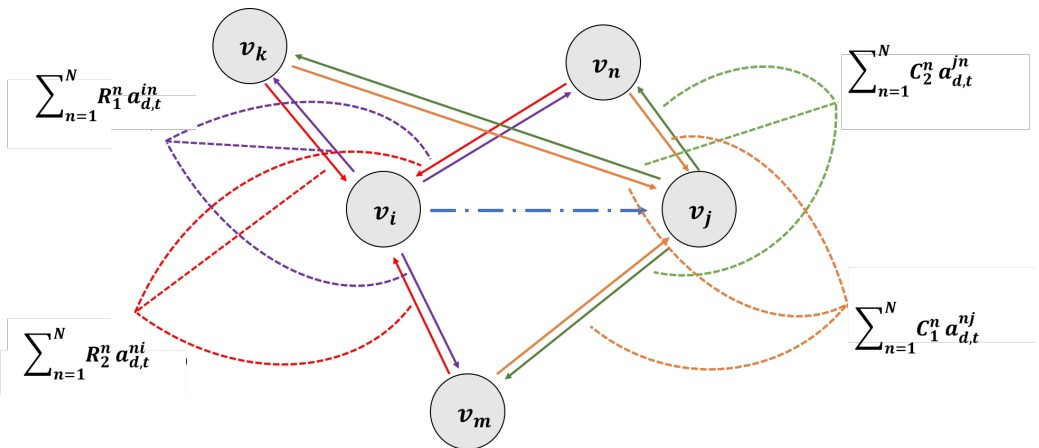

Figure 5: Working mechanism of spatial adjacent convolution network (VACN) for a target OD flow from $v_i$ to $v_j$

## C    TRAINING DETAILS

Batch normalization is used in the VACN component. The batch size in our experiment was set to 10, corresponding to 10 randomly sampled timestamps and all the $50^2$ OD flows in each snapshot. The initial liearning rate is set to be $10^{-4}$ with a decay rate $10^{-6}$. We use early stopping for all the deep learning-based methods where the training process is terminated when the RMSE over validation set has not been improved for 10 successive epochs. The maximal number of epochs allowed is 100.

## D    TABLE3: COMPARISON WITH STATE-OF-THE-ART METHODS FOR CITY B

## E    COMPARISON OF DIFFERENT HYPERPARAMETER CONFIGURATIONS

In this section, we want to explore how some important hyperparameters of input OD flow data, for example $p_1$ and $p_2$, may affect the performance of our LSTOD model.

Table 3: Comparison with State-of-the-art methods for City B

| Method | City B 30 min | 60 min | 90 min |
|---|---|---|---|
| HA | | 3.95(2.78/4.58/2.21) | |
| ARIMA | 4.87(3.81/6.83/1.22) | 5.14(3.91/6.34/1.71) | 5.39(4.04/6.65/2.15) |
| LSVR | 4.21(3.19/5.36/1.98) | 4.87(3.56/6.29/3.06) | 5.24(3.77/6.83/3.68) |
| LSM-RN | 5.99(5.12/7.05/2.03) | 6.72(5.70/7.74/2.14) | 7.32(6.30/8.35/2.34) |
| DLSTM | 3.86(3.34/4.37/0.59) | 4.04(3.52/4.60/0.65) | 4.52(3.89/5.11/0.81) |
| SRCN | 2.89(2.33/3.41/0.59) | 3.06(2.53/3.70/0.80) | 3.17(2.50/3.87/1.02) |
| **LSTOD** | **2.56(2.25/2.85/0.20)** | **2.63(2.29/2.95/0.24)** | **2.69(2.32/3.05/0.31)** |

Figure 6 (b) compares RMSE on testing data by STOD model with different data settings. Varied combinations of the short-term sequence length $p_1$ and the long-term day-level sequence length $p_2$ are studied. We can see that the best performance is achieved as $(p_1, p_2) = (7, 5)$ with RMSE = 2.41. Specifically, settings with different $p_1$'s under $p_2 = 5$ consistently outperform those under $p_2 = 7$. It may demonstrate that the shift can usually be captured within a short time range, while a longer time sequence may smooth the significance. Table 4 provides the detailed prediction results for each data setting.

Table 4: Comparison of STOD under different $p_1$, $p_2$ combinations

| $p_2$ | $(K_{LT}^0, K_{LT}^1)$ | $p_1$ | $(K_{ST}^0, K_{ST}^1)$ | RMSE |
|---|---|---|---|---|
| | | 5 | $(2, 2)$ | 2.45 |
| | | **7** | $(\mathbf{2, 3})$ | **2.41** |
| 5 | $(2, 2)$ | 9 | $(3, 3)$ | 2.42 |
| | | 11 | $(3, 4)$ | 2.43 |
| | | 13 | $(4, 4)$ | 2.43 |
| | | 5 | $(2, 2)$ | 2.45 |
| | | 7 | $(2, 3)$ | 2.44 |
| 7 | $(3, 2)$ | 9 | $(3, 3)$ | 2.44 |
| | | 11 | $(3, 4)$ | 2.44 |
| | | 13 | $(4, 4)$ | 2.49 |

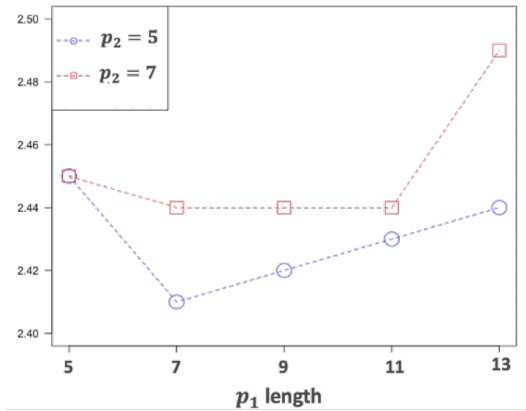

Figure 6: RMSE on testing data with respect to STOD with different $p_1$ and $p_2$ combinations.

