# OpenReview forum: "LSTOD: Latent Spatial-Temporal Origin-Destination prediction model and its applications in ride-sharing platforms"
_ICLR.cc/2020/Conference — Reject_

### Official Review · AnonReviewer2 · 2019-10-22
**Official Blind Review #2**

**Rating:** 1

**Review:**

The paper proposes a  deep learning architecture for forecasting Origin-Destination (OD) flow. The model integrates several existing modules including spatiotemporal graph convolution and periodically shifted attention mechanism. Experiments on a large-scale ride-sharing demand dataset demonstrate improved forecasting accuracy.

+ The spatiotemporal forecasting problem is an important task in the context of ride-sharing platforms.
+ The description of the model is clear and the real-world dataset experiments are interesting.

- The novelty of the model is rather limited. Spatiotemporal graph convolution has been previously proposed in [Yu et al. 2018] and later studies. Periodically shifted attention is a slight modification of the earlier work.
- The mathematical notations are messy and redundant. For example, the superscript index i, j can be removed and write all equations in matrix form. The subscript index can be replaced with a range operator. In Eqn (2), r_1 and c_2 are non-identifiable.
- The experiments are not convincing. The paper is missing the following important baselines:

Bing Yu, Haoteng Yin, and Zhanxing Zhu. Spatio-temporal graph convolutional networks: A deep learning framework for traffic forecasting. In Proceedings of the 27th International Joint Conference on Artificial Intelligence, 3634-3640. AAAI Press, 2018.

Yaguang Li, Rose Yu, Cyrus Shahabi, Yan Liu. "Diffusion Convolutional Recurrent Neural Network: Data-Driven Traffic Forecasting." International Conference on Learning Representations (ICLR), 2018.

The authors should provide confidence intervals and visualizations for the predictions.




**Experience Assessment:**

I have published in this field for several years.

**Review Assessment: Checking Correctness Of Derivations And Theory:**

I carefully checked the derivations and theory.

**Review Assessment: Checking Correctness Of Experiments:**

I carefully checked the experiments.

**Review Assessment: Thoroughness In Paper Reading:**

I read the paper thoroughly.

---

> ### Author Response · Authors · 2019-11-15
> **Response to Official Blind Review #2**
>
> We want to appreciate many insightful comments from this reviewer first. However, there may be some misunderstanding we want to clarify.
>
> Sorry for the confusion of our VACN layer with the traditional graph convolutions. We have included some explanations in the introduction section (marked by color blue) to elaborate the key difference between VACN and GNNs. Both the observations and responses for the prediction of graph convolutions are vertex-level variables. On the contrary, the OD flows that we discuss in this paper are generated in the edge space according to our definition, and this is why we introduce a novel CNN architecture ‘VACN’ to model the adjacency information between edges in a graph. This is the reason that the models proposed in Yu et.al. and Li et.al. are not applicable here.
>
> To better understand the differences between our proposed VACN over edges of a graph with those convolutions over vertices of a graph such as GCN, we can use the concept of line graphs L(G) with each node in L(G) corresponding to an edge in G. Each individual edge in $L(G)$ can be mapped to a pair of edges in G that share a common vertex. L(G) contains 1/2 [N^3 - N(N-1)], while there are only 1/2N(N-1) in G. Thus, applying GNNs over $L(G)$ is memory intensive. Our edge-based convolution operations VACN is much more efficient.
>
> On the other hand, we have included mean, variance and confidence intervals for all the experiments and compared different models. We have also included new plots such as Figure 3 in order to demonstrate how the complete LSTOD model outperforms its variants from both local and global perspectives.

---

### Official Review · AnonReviewer1 · 2019-10-22
**Official Blind Review #1**

**Rating:** 6

**Review:**

Summary: This paper proposes a latent spatial-temporal origin-destination model to address the OD flow prediction problem.

The main contributions are summarized as follows:

1.	The authors propose a purely convolutional framework to learn both short-term and long-term spatio-temporal features simultaneously from dynamic origin-destination flow data.
2.	The authors propose a novel SACN architecture to capture the relevance of OD flows by modeling each OD flow map as an adjacency matrix.
3.	The authors design a periodically shift attention mechanism to model the long-term periodicity.
4.	The results demonstrate that the proposed model outperforms state-of-the-art methods in OD flow prediction, with 6.5% to 15.0% improvement of testing RMSE.

However, my major concerns are as follows:

1. On Page 4, the authors explain the reasons why they use TGCNN instead of RNN-based architectures to capture the temporal representation. However, it would be more convincing if quantitative analysis or empirical results are provided.

2. On Page 4, readers might be confused with the symbols in the formulation (3) that are not defined clearly.

3. On Page 7, the experiment only considers one metric, i.e., RMSE. The efficiency comparisons between the proposed model and the baselines are missing.

**Experience Assessment:**

I have read many papers in this area.

**Review Assessment: Checking Correctness Of Derivations And Theory:**

I assessed the sensibility of the derivations and theory.

**Review Assessment: Checking Correctness Of Experiments:**

I carefully checked the experiments.

**Review Assessment: Thoroughness In Paper Reading:**

I read the paper at least twice and used my best judgement in assessing the paper.

---

> ### Author Response · Authors · 2019-11-15
> **Response to Official Blind Review #1**
>
> We appreciate many insightful comments from this reviewer. We have rewritten all the formulations in Section 3, and the definitions now are much clearer. All the revised parts are marked in red. For the efficiency, we currently have limited access to the raw data, so we cannot re-run all the experiments to compare the computation efficiency in a short period. On the other hand, we have included more experiment results other than RMSE (mean, variance, percentiles) and some visualization plots in order to demonstrate the advantage of our architecture over the baselines.

---

### Official Review · AnonReviewer3 · 2019-10-23
**Official Blind Review #3**

**Rating:** 1

**Review:**


Review Summary
--------------
While I think there are some interesting innovations here, I don't think this is ready for ICLR. I have concerns that the evaluation doesn't focus enough on application-relevant scenarios (why exclude cells that are not in the top 50? why not use 7-day windows to capture day-of-week effects?), and that the evaluations don't quantify uncertainty (and thus the claimed improvements due to attention may not be significant). The presentation quality of the method details in Sec. 3 needs a significant rewrite to improve clarity. I do think the non-standard convolution operator is a nice idea.

Paper Summary
-------------
This paper addresses the forecasting of origin-destination demand data, with a focus on ride-sharing transportation applications. We assume an urban area has been divided up into N cells (known in advance). The problem is to forecast demand for *directed* rides from cell i to cell j, using historical demand data.

The paper's first contribution is the design of a convolutional neural net architecture that uses non-rectangular receptive fields more suitable to origin-destination data. Instead of assuming that cells in a given 2D euclidean neighborhood are correlated, it assumes that all journeys that overlap with the one of the cells in the current origin-destination pair are similar. They call this the "SACN" architecture.

The second contribution is developing an "ST-Conv" block that combines the above SACN convolution with gated temporal convolutions, so that both space and time can be summarized via convolution operators. A periodically-shifting attention mechanism is further used to measure long-term data's similarities to short-term data and weight accordingly.

Overall, these two ideas (SACN convolutions and ST-Conv blocks with attention) are combined into what they call their Latent Spatio-Temporal Origin-Destination or "LSTOD" architecture, which is claimed to be the first to use both short-term and long-term features in prediction.

Evaluation examines demand for ride-sharing in two major cities, A and B (presumably masked because they are proprietary). RMSE error comparing to several classic (e.g. ARIMA) and deep feature learning (e.g. LSTMs and SRCN) baselines, with primary results in Table 1. A few side experiments examine superiority over standard convolutions (Fig 3) and experiments without the attention and long-term bits of the model (Table 2).


Novelty & Significance
-----------------------
The current paper is quite niche when it comes to significance; I think it may be a bit too specialized for most ICLR readers. Solving this kind of forecasting problem seems important to ride-sharing applications, but is highly specialized for origin-destination demand data. Would be nice to see the paper attempt to connect to other problems beyond ride-sharing (maybe animal migration? maybe package logistics?).

That said, I think there's sufficient novelty. The architectural design contributions here do appear new to me (though I don't follow this kind of data closely).


Method Concerns
------------------

## M1: Complexity analysis missing, scaling could be a problem

When comparing the current square receptive field architecture for CNNs with the proposed SACN, I felt there was an opportunity to clarify how both scale with N and other key problem size parameters in terms of number of parameters or execution time. I'm concerned that it will be non-scalable given the O(N) cost in terms of number of parameters and the O(N^2) cost of runtime (you need to execute Eq. 2 for each of the N^2 entries at each layer). I'd like to see some careful breakdown of this compared to other approaches.

Experimental Concerns
---------------------

## E1: Why not use a 7-day past history?

Shouldn't ride share demand have a day-of-week trend? Why wouldn't we use last Sunday's demand to predict this Sunday's demand? I find it quite odd that for the Historical Average baseline, only the last 5 days (rather than last 7) are used, and for the presented method, only the last 3 days are used. I'd be happy to be proven wrong, but I'd guess including a 7-day window would lead to noticeably better results.

## E2: Lack of error bars / uncertainty quantification

A natural question is, can we reliably tell that the difference between (for example) RMSE 2.49 and 2.54 is significant and not noise?  I'd like to see some attempt at quantifying the uncertainty for measurements in Table 1 (perhaps taking each full day in test set as its own "mini" test set that produces one RMSE score, then reporting average as well as 2.5th and 97.5th percentiles or something across each day). Without this, I think the claim that attention is useful here is unproven, since the change in performance is so small (less than 0.1 RMSE).

## E3: Why focus only on the N=50 most common cells?

I would think to really assess demand forecasting, you want to know when to task drivers to visit less-common cells. The focus on the cells with only the top 80% of journeys means that 1 of every 5 rides would not be covered by this prediction system. I wonder if part of the reason is that using N much larger than 50 is problematic due to the scaling mentioned earlier.


Presentation Concerns
---------------------

## P1: Need to simplify notation

I found the descriptions of the neural net architecture throughout Sec. 3 quite hard to parse. I think there's an overreliance on math notation and there could be more simple description of high level intent and motivation.

For example, Eq. 2 could be simplified to avoid the channel "m" and layer "l" notation and thus let the reader focus on what matters, which is how any part of input A that involves the origin node i or the destination node j is included in the weighted sum.  Words can be used around this to clarify this same operation can happen across channels and layers.

Similarly, Eq. 3 and Eq. 4 could perhaps be replaced by a good diagram. Currently, Eq. 3 both P and Q are undefined and quite confusing.

## P2: Name of the convolution operation

I'm not sure "SACN" is the best name. The receptive field of a standard CNN looks much more "spatially adjacent" to me (a compact rectangle surrounding the target cell). Instead, you might call it "topologically adjacent" or even something like "vertex adjacent". You want to emphasize that you are getting strength by finding all journeys whose start or end overlaps with one of the endpoints of the current journey.

**Experience Assessment:**

I have read many papers in this area.

**Review Assessment: Checking Correctness Of Derivations And Theory:**

N/A

**Review Assessment: Checking Correctness Of Experiments:**

I assessed the sensibility of the experiments.

**Review Assessment: Thoroughness In Paper Reading:**

I read the paper at least twice and used my best judgement in assessing the paper.

---

> ### Author Response · Authors · 2019-11-15
> **Response to Official Blind Review #3**
>
> We appreciate many insightful comments from this reviewer. Below are some responses to your concerns.
>
> ## P1: Need to simplify notation
> We have removed the "m" and layer "l" in Eq2. Moreover, we have rewritten Section 3.1 to more clearly clarify our proposed spatial-level operator and how the connected OD flows contribute to the output representation of the target OD flow. We have also rewritten Sections 3.2 – 3.4 to simplify notation and clarify various definitions. Please refer to the red parts in Sections 3.1-3.4 for more details.
>
> ## P2: Name of the convolution operation
> Thanks a lot! We have replaced the name ‘SACN’ by ‘VACN’ to emphasize the adjacency between OD flows that share common vertices in the graph.
>
> ## M1: Complexity analysis missing, scaling could be a problem
> Yes, the computational complexity of our proposed VACN is O(N). However, both VACN and standard local CNN are applied to all the N^2 entries in this case. Therefore, when the side length of the CNN filter is in the order O(N^0.5), the total costs of our VACN and standard CNN are also in the same order. Please refer the last paragraph in ‘ACN VS standard local CNN’ subsection of Section 4.4
>
> ## E1: Why not use a 7-day past history?
> Thanks a lot! We have now used the same day in the previous 4 weeks for Historical Average instead of previous 5 days. The reduced RMSEs demonstrate that there exists significant long-term periodicity.
>
> ## E2: Lack of error bars / uncertainty quantification
> As suggested, we have added the mean, variance, 25% quantile and 75% quantile of the 14 day-wise RMSEs on the testing set and the results are summarized in Tables 1 and 2. We have also included more explanations to these new results, which are marked in red in Section 4. Moreover, we do not use 2.5% and 97.5% here since there are only 14 observations in each sample. We have also added some new plots such as Figure 3 to show how the complete LSTOD model outperforms its variants from both local and global perspectives. More details can be found in Section 4 and Figure 3.
>
> ## E3: Why focus only on the N=50 most common cells?
> The main reason is that the computation complexity and memory cost highly increases as N gets larger since all the compared CNN operators need to be applied N^2 times at each timestamp. On the other hand, the N^2 = 2500 OD flows can cover a sufficient proportion of the total demands (more than 80%) to satisfy the operation requirement of the ride-sharing platform.

---

### Decision · Program_Chairs · 2019-12-19

**Decision:**

Reject

**Comment:**

The paper proposes a  deep learning architecture for forecasting Origin-Destination (OD) flow. The model integrates several existing modules including spatiotemporal graph convolution and periodically shifted attention mechanism.

The reviewers agree that the paper is not written well, and the experiments are also not executed well. Overall, we recommend rejection.